



Title page

## 2    Effect of colloidal particle size on physicochemical properties

## 3    and aggregation behaviors of two alkaline soils

Yu-yang Yan[1], Xin-ran Zhang[1], Chen-yang Xu[1,2]*, Jun-jun Liu[1], Fei-nan
Hu[3,4], Zeng-chao Geng[1,2]
(1. *College of Natural Resources and Environment*, *Northwest A&F University*,
*Yangling, Shaanxi* 712100, *China*; 2. *Key Laboratory of Plant Nutrition and the Agri-*
*environment in Northwest China, Ministry of Agriculture, Northwest A&F University,*
*Yangling, Shaanxi 712100, China; 3. State Key Laboratory of Soil Erosion and Dryland*
*Farming on the Loess Plateau, Northwest A&F University, Yangling, Shaanxi 712100,*
*China; 4. Institute of Soil and Water Conservation, Chinese Academy of Sciences,*
*Ministry of Water Resources, Yangling, Shaanxi 712100, China*)
**\*Corresponding author:**
Chen-yang Xu
Email address: xuchenyang@nwafu.edu.cn, xuchenyang.ms@163.com;
Postal address: College of Natural Resources and Environment, Northwest
A&F University, No. 3 Taicheng Road, Yangling District, Shaanxi 712100,
China.



**Abstract**
Soil colloidal particles are the most active components of all, and they also vary in
elemental composition and environmental behaviors with the particle size. The
purposes of the present study are to clarify how particle size affects the physiochemical
properties and aggregation kinetics of soil colloids, and to further reveal the underlying
mechanisms. Soil colloidal particles from two alkaline soils—Lou soil and cinnamon
soil were subdivided into three ranges: $d < 2$ μm, $d < 1$ μm and $d < 100$ nm. The organic
and inorganic carbon contents, clay mineralogy, surface electrochemical properties,
including surface functional groups and zeta potentials, were characterized. Through
time-resolved light scattering technique, the aggregation kinetics of soil colloidal
fractions were investigated, and their critical coagulation concentrations (CCCs) were
determined. With decreasing colloidal particle diameter, the total carbon content,
organic carbon, organic functional groups content and illite content all increased. The
absolute zeta potential values and the charge variability decreased with decreasing
particle diameter. The CCC values of Lou soil and cinnamon soil colloids followed the
descending order of $d < 100$ nm, $d < 1$ μm, $d < 2$ μm. Compared with the course factions
($d < 1$ μm and $d < 2$ μm), soil nanoparticles were more abundant in organic carbon and
more stable clay minerals ($d < 100$ nm), thus they exhibited strongest colloidal
suspension stability. The differences in organic matter contents and clay mineralogy are
the fundamental reasons for the differences in colloidal suspension stability behind the
size effects of Lou soil and cinnamon soil colloids. The present study revealed the size
effects of two alkaline soil colloids on carbon content, clay minerals, surface properties



and suspension stability, emphasizing that soil nanoparticles are prone to be more stably
dispersed instead of being aggregated. These findings can provide references for in-
depth understanding of the environmental behaviors of the heterogeneous soil organic-
mineral complexes.
**Keywords:** Nanoparticles; Clay mineral composition; zeta potential; Critical
coagulation concentration



## 1. Introduction


Soils contain a series of solid particles in continuous sizes, ranging over six orders of
magnitude from nanometers to millimeters (Lead and Wilkinson, 2006; Li et al., 2011),
among which soil colloids are the most active parts. Soil colloids are characterized by
high surface area and strong adsorption capacity, which can largely determine the fate
and transport of pathogens, nutrients, heavy metals and organic pollutants, and might
cause environmental problems to adjacent water bodies or groundwater (Baalousha et
al., 2009). Due to their high reactivity and fluidity in aqueous environment, soil colloids
play an important role in physical, chemical and biogeochemical processes of natural
environment (Schäfer et al., 2012; Mayordomo et al., 2016). The capacity of soil
colloids in driving attached nutrients and pollutants is closely related to their dispersion
stability under various environmental conditions (Won and Burns, 2018). Therefore,
studies on the dispersion stability of soil colloids have attracted extensive attention.
Currently, the definition of soil and environmental colloidal fractions is ambiguous.
Soil colloidal fractions are defined as soil particles in diameter of < 1 μm (Lead and
Wilkinson, 2006; Weil and Brady, 2016), and also being of < 2 μm (Zhang et al., 2021);
while in some extreme cases, they can refer to the particles in diameter of 5–10 μm (Yin
et al., 2010). Such discrepancies are seen among publications due to the fact that
colloids are defined based on the particle diameter range within which they can display
colloidal properties. Since for different materials, e.g., metal (Fe/Al/Ti) oxides, silica
gel, phyllosilicates, the specific colloidal range differs greatly.
Compared with engineered nanoparticles with known mineralogical organization,





natural soils are much more heterogeneous (Cárdenas et al., 2010); their elemental
composition and clay mineralogy of soil colloids change with particle size. Tsao et al.,
(2013) found that quartz and feldspar were mainly dominant in colloidal particles of <
2 μm and 450–2000 nm in red soil, while illite and montmorillonite were the main clay
minerals in nanoparticles (1–100 nm). In addition, the mineral structure at nanometer
scale also changes. Compared with colloidal particles of < 2 μm, the Si/Al ratio in
nanoparticles increased, and the surface area, morphology, crystallinity, surface atomic
structure and frame structure were significantly different (Tsao et al., 2011).
Furthermore, particle size also affects the surface potential of soil colloids. Tang et al.,
(2015) investigated the surface potential variations with particle size (1–10 μm, 0.5–1
μm, 0.2–0.5 μm, < 0.2 μm) for variably-charged yellow soil and permanently-charged
purple soil; among the colloidal fractions, the absolute surface potential of the finest
particles of purple soil was lowest while that of the yellow soil was the largest, caused
by the differences in surface charge density. Thus, the influences of particle size on
elemental composition and surface properties of soils should be further studied.

In recent years, great progress has been made in the study of dispersion stability

of soil clay minerals, such as montmorillonite, kaolinite, illite or hematite, and soil
nanoparticles (Xu et al., 2018; Sun et al., 2020; Wei et al., 2021; Zhu et al., 2014). He
et al., (2008) demonstrated that hematite nanoparticles with various particle diameters
showed different surface properties and aggregation behaviors under the same pH
conditions; moreover, the critical coagulation concentrations (CCCs) of hematite
decreased with the decrease of particle diameter. Zhou et al., (2013) compared the



CCCs of ten different $TiO_2$ nanoparticles with varying sizes and indicated that crystal
structure and particle diameter both affected the aggregation behaviors of $TiO_2$. Zhang
et al. (2016) confirmed that the types of clay minerals for two Alfisols changed from
smectite and vermiculite to kaolinite and illite when the particle size varied from
colloids to nanoparticles. Therefore, the dependence of physiochemical properties,
surface properties and environmental behaviors on particle size for heterogeneous soil
colloidal particles needs systematic investigation.
In the present study, soil colloidal particles of two alkaline soils—Lou soil and
cinnamon soil were subdivided into three ranges: $d < 2$ μm, $d < 1$ μm and $d < 100$ nm.
Their organic fraction and clay mineralogy, surface electrochemical properties and
colloidal stability were studied. The objectives of the present study are to clarify how
particle size affects the surface properties and aggregation behaviors of soil colloids,
and to analyze the underlying mechanisms. The findings are of important significances
for predicting the environmental performances of colloids and colloid-facilitated
nutrients, pollutes and pathogens in natural soil and water environment.

## 2. Materials and methods

### 2.1 Soil sampling

Two representative surface soils (0–20 cm) were collected from the Guanzhong
Plain, China, namely Lou soil (agricultural soil) and cinnamon soil (natural soil). Lou
soil was collected from Yangling District, Shaanxi Province. Cinnamon soil was
collected from Zhouzhi County, Shaanxi Province. Lou soil and cinnamon soil are
classified as Anthrosols and Calcisols, respectively, according to the FAO soil



classification. Soils were taken back to laboratory for air-drying and sieving. The basic
soil properties are determined based on standard methods. The pH of Lou soil was 8.34
while it was 8.32 for the cinnamon soil. The organic carbon contents of Lou soil and
cinnamon soil were 7.25 g·kg$^{-1}$ and 9.22 g·kg$^{-1}$, respectively. The contents of $CaCO_3$
in Lou soil and cinnamon soil were 51.7 g·kg$^{-1}$ and 82.5 g·kg$^{-1}$. The Free Fe/Al oxides
content of Lou soil and cinnamon soil were 22.8 g·kg$^{-1}$ and 23.1 g·kg$^{-1}$.
**2.2 Extraction of soil colloidal fractions in different size ranges**
The soil colloidal particles were extracted based on the Stokes' law, and the
detailed procedures can be found in our previous publication (Hu et al., 2022). Briefly,
weigh 50 g soil into a beaker containing 500 mL distilled water, and put the suspension
under sonication for an hour using the ultrasonic cell disrupter (XO-900D, Nanjing
Xianou Instruments Corporation, China) while maintaining the temperature below
30°C. Afterwards, the suspension was transferred to a larger beaker and distilled water
was added to make up the total volume of 5 L. The suspension was further dispersed
using an electronic blade stirrer (JB-200, Shanghai Nanhui Huiming Apparatus, China)
for one hour, before being sifted through a 300-mesh sieve, and the upper suspensions
containing soil colloidal particles in different diameters were collected by
centrifugation. Based on the equation (1), centrifugation speed and time for colloidal
particles of $d < 2$ μm, $< 1$ μm and $< 100$ nm are calculated and shown in Table S1.

$$t = \frac{\eta \lg \frac{R_2}{R_1}}{3.81 N^2 r^2 \Delta d}$$

(1)

in which, $t$ is time for centrifugation (s); $R_1$ is the distance from the surface of the liquid





to the center of the axis of the centrifuge, here is 5.7 cm; $R_2$ is distance from the particles
to the center of the axis of the centrifuge, here is 10.5 cm; $N$ (rev·s$^{-1}$) is the centrifuge
speed; $r$ (cm) is the desired colloidal particle radius; $\Delta d$ is the difference in density
between the soil particles and water, while $\Delta d$ is 1.65 g·cm$^{-3}$; $\eta$ is the water viscosity
coefficient, here is 0.00839 g·cm$^{-1}$·s$^{-1}$ at 25 °C.

**2.3  Characterization of soil colloidal fractions in different size ranges**

The initial particle diameters of soil colloids were determined by a time-resolved

dynamic light scattering (DLS) apparatus (Nanobrook Omni, Brookhaven, USA). The
organic carbon contents in soil colloids were determined by potassium dichromate
external heating method and total carbon content was determined by elemental analyzer
(Elementar Vario EL III, Germany). The inorganic carbon content was calculated by
subtraction method (Wang et al., 2011). The clay mineralogy of soil colloids was
determined by the XRD (Ultima-IV, Rigaku, Japan). The specific surface areas of the
soil colloids were measured by BET-N$_2$ method (ASAP 2460, Micromeritics
instrument, USA). High-resolution spectra of C1s and O1s of soil colloids were
acquired by X-ray photoelectron spectroscopy (XPS) (Thermo Scientific K-Alpha,
USA) (Luo et al., 2019). The zeta potentials of soil colloids were measured by Zeta
PALS equipped with a BI-ZTU Autotitrator (ZetaPALS, Brookhaven, USA) with 1
mmol·L$^{-1}$ NaCl solution as the background electrolyte; and the pH range of colloidal
suspension was set to 3–10 adjusted with 0.1 mol·L$^{-1}$ HCl and NaOH. The
concentrations of K$^+$, Na$^+$, Ca$^{2+}$, and Mg$^{2+}$ in soil colloidal suspensions were measured
by flame atomic absorption spectrophotometry (PinAAciie 900F, USA).





**2.4 Aggregation kinetics of soil colloidal fractions**

The aggregation kinetic curves of soil colloidal particles in different electrolytes were determined by time-resolved DLS measurements. The incident wavelength was 635 nm and the scattering angle was 90°. While using the DLS instrument, it is necessary to clean up the surrounding dust, especially the sample pool. The stock colloidal suspensions with particle concentration of 200 mg·L$^{-1}$ were mixed with electrolyte solutions with equal volume. The suspension pH was adjusted to 8.0, which was close to the pH value of natural soil with addition of 0.1 mol·L$^{-1}$ HCl or NaOH before measurement. The chosen electrolyte concentrations for NaCl and CaCl$_2$ were 200–2000 and 2–20 mmol·L$^{-1}$. The effective diameter ($D_h$) of the mixed sample was automatically recorded every 2 min, and an aggregation kinetic curve was obtained in 30 min monitoring.

**2.5 Calculation of critical coagulation concentration**

According to the particle interaction theory, the aggregation kinetic curves under electrolyte conditions can be divided into reaction-limited aggregation (RLA) stage under low concentration which was affected by electrolyte conditions and diffusion-limited aggregation (DLA) stage under high concentration which was not affected by electrolyte concentration. The CCC is the critical electrolyte concentration when the aggregation process changes from the RLA state ($\alpha < 1$) to the DLA state ($\alpha = 1$). Attachment efficiency ($\alpha$) represents the bonding probability of particle collisions and can be calculated for each electrolyte concentration by using equation 2, which allowed the curve of $\alpha$ as a function of electrolyte concentration to be plotted (Xu et al., 2020a;



Hu et al., 2022).

$$\alpha_{\exp} = \frac{1}{W} = \frac{k_{11}}{\left(k_{11}\right)_{fast}} = \frac{\dfrac{1}{N_0}\left(\dfrac{da_h(t)}{dt}\right)_{t\to 0}}{\dfrac{1}{\left(N_0\right)_{fast}}\left(\dfrac{da_h(t)}{d_t}\right)_{t\to 0,\,fast}}$$     (2)

where $D_h$ is the effective diameter of particles, $t$ is the time (min); $N_0$ is the density of
particles; $K_{11}$ is the aggregation rate of RLA; $(K_{11})_{fast}$ is the aggregation rate of DLA.
The intersection of RLA regime and DLA regime is the CCC.

The aggregation rates were calculated by the average of the last 5 effective

diameters divided by the aggregation time at specific electrolyte concentration. The
fractal dimension in the DLA regime was obtained based on the method proposed by
Wang et al. (2013).

$$D(t) = b * t^n + D_0$$     (3)

in which, $D(t)$ is the colloidal effective diameter at time t (min), $D_0$ is the initial effective
diameter of colloids, and $b$ and $n$ are constants determined by the aggregation curves.
The fractal dimension is $d_f = 1/n$ in the DLA regime.
**3. Results and discussion**
**3.1 Particle size and distribution characteristics of Lou soil and cinnamon soil**

**colloidal fractions**

The average diameters of Lou soil and cinnamon soil colloids were measured by

time-resolved DLS, and the results were shown in Table 1. The number-weighted
diameters for Lou soil colloids of $d < 2$ μm, $< 1$ μm and $< 100$ nm were 133.16, 127.84
nm and 72.47 nm, respectively. The intensity-weighted diameters were 396.81 nm,



371.45 nm and 294.10 nm. For cinnamon soil colloidal fractions, the number-weighted
diameters for colloids of $d < 2$ μm, $< 1$ μm and $< 100$ nm were 141.23 nm, 131.67nm
and 85.48 nm, and their intensity-weighted diameters were 439.20 nm, 372.07nm and
312.25 nm, respectively. The intensity-weighted diameters were generally higher than
the number-weighted diameters, especially in polydisperse system (Xu et al., 2020b).
The particles in the soil solution are in constant Brownian motion, and when light passes
through the colloids, the particles will undergo light scattering, resulting in fluctuations
in light intensity, and thus the effective diameter (intensity-weighted diameter) of the
particles is calculated (Filella et al., 1997). Since the particle diameter is proportional
to the sixth power of the light intensity, that is, if there are larger particles in the solution
in such polydisperse systems, the number-weighted diameter is generally more
representative of the true diameter of colloidal particles (Xu et al., 2015).
*(Insert Table 1 near here)*
From table 1, it can be seen that the average colloidal diameters of $d < 2$ μm were
close to that of $d < 1$ μm, and they were both significantly higher that of the nano-sized
fraction. From the particle size distribution characteristics, it is clear that the size range
indicated by the differences of $D_{90}$ and $D_{10}$ increased with intended particle diameter.
For Lou soil and cinnamon soil, 74.69% and 63.55% of all particles contained in the
colloidal suspensions of $d < 100$ nm were actually less than 100 nm, respectively,
indicating the complexity of soil colloidal particle irregularity.

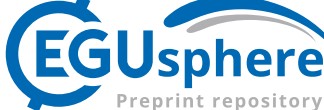

### 3.2 Physiochemical properties and clay mineralogy of Lou soil and cinnamon soil colloids

Table 2 shows the physiochemical properties of soil colloidal fractions. The yields
of each colloidal fraction of Lou soil were slightly larger than that of cinnamon soil,
respectively. The yields of colloidal particles of $d < 2$ μm were about 1.3~1.4 times of
$d < 1$ μm, and about 4.0~4.9 times of $d < 100$ nm, respectively. With the decreasing
colloidal particle diameter, the total carbon content, organic carbon and inorganic
carbon content all increased, suggesting the finer particles were richer in carbon. This
tendency is in agreement with other publications (Zhang et al., 2021; Said-Pullicino et
al., 2021; Hu et al., 2022). The specific surface areas for colloidal fractions of $d < 1$ μm
were largest of all, which may be related to the structures of formed clusters while
drying the samples for observation under microscopy (Yu et al., 2017; Weissenberger
et al., 2021).

*(Insert Table 2 near here)*

The clay mineralogy of Lou soil and cinnamon colloidal fractions is shown in
Table 3. Cinnamon soil colloidal fractions were dominant by illite, kaolinite and
chlorite while there was less chlorite in Lou soil colloidal fractions. With the decrease
of particle size, the content of illite increased and kaolinite content decreased. This
tendency is in agreement with other publications (Chenu and Plante, 2006; Zhang et al.,
2016). Among the dominant clay types, the size of illite is finer than kaolinite and
chlorite (Weil and Brady, 2016), so its mass percentage was higher in the nano-sized
fraction.



*(Insert Table 3 near here)*

### 3.3 Surface properties of Lou soil and cinnamon soil colloids

The XPS spectra of soil colloidal fractions are shown in Fig. 1. From Fig. 1, it can
be seen that the main C-containing functional groups were C–C/C-H/C=C, C–O, C=O,
and COO- groups at 284.6, 286.2, 288.0 and 289.4 eV, respectively (Liang et al., 2020;
Ding et al., 2023). The functional groups for colloidal particles of $d < 100$ nm were
more abundant than those for colloids of $d < 2$ μm and $d < 1$ μm, while there were no
significant differences between colloids of $d < 2$ μm and $d < 1$ μm. With the decrease
of colloidal particle diameter, the relative contents of oxygen-containing functional
groups (C–O, C=O, COO-) gradually decreased. Specifically, the content decreased
gradually from 32.01% in Lou colloids of $d < 2$ μm to 20.93% in Lou colloids of $d <$
100 nm (Table S2). The functional groups of C–O and COO- gradually decreased until
they eventually disappeared, more C=O groups were exposed to the surrounding air.
For cinnamon soil colloids (Fig. 1d, e, f), the relative contents of organic oxygen-
containing functional groups for colloidal particles of $d < 2$ μm, $d < 1$ μm and $d < 100$
nm showed a different trend, compared with that in Lou soil colloids. The relative
contents of organic oxygen-containing functional groups gradually increased with the
decrease of diameter. This trend was particularly pronounced in fraction of $d < 100$ nm,
and contents of C–O and COO- were highly increased (Table S2).
*(Insert Figure 1 near here)*
Oxygen-containing functional groups of C–O, C=O and COO- are electronegative
functional groups, hydroxyl and carboxyl groups can lose protons and make the surface
of soil colloidal particles carry negative charges (Audette et al., 2021). Functional
groups of C–O, C=O and COO- can affect the negative charges carried on the colloidal



surface by forming hydrogen bonds, and their polarity can also affect the negative
charges on the surface when O atom combines with C and H. The electrons will lean
towards the O atom with stronger electronegativity, which also makes the colloidal
surface carry negative charges (Tan et al., 2019). The contents and types of oxygen-
containing functional groups are one of the main factors affecting colloid charge and
aggregation.

The zeta potential values of different colloidal fractions at the pH range of 3–10

are shown in Fig. 2. Zeta potentials of the colloidal particles were negative, indicating
that they were negatively-charged. The absolute values of zeta potentials for of Lou soil
and cinnamon soil colloidal particles increased with increasing solution pH, due to the
deprotonation of the surface (Moayedi and Kazemian, 2013; Dong et al., 2019).
Compared with the cinnamon soil colloids, the zeta potentials of Lou soil colloidal
particles were more negative. For cinnamon soil colloids, the differences among
colloidal fractions were larger.

(*Insert Figure 2 near here*)

In general, the absolute zeta potential values increased with increasing particle

diameter. When the pH changed from 3 to 10, for every pH unit increased, the zeta
potential values of Lou soil colloids of $d$ < 2 μm, < 1 μm, and < 100 nm would be
increased by 2.14 mV, 2.09 mV and 1.89 mV; and for cinnamon soil colloids, those
variation rates were 2.15 mV, 1.45 mV and 1.37 mV, respectively. Those data
demonstrate that the charge variability decreasing with the decreasing particle diameter.
Song et al., (2019) compared the zeta potential of wheat straw biochar nanoparticles (<



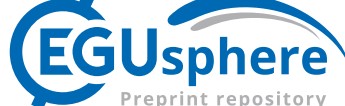

100 nm) and colloidal particles (< 1000 nm), and found that the absolute values of
colloidal particles were larger at same pH, which was explained by the differences in
the number of surface carboxyl and hydroxyl groups. The zeta potential of colloidal
particles is proportional to charge density, which means that it is related to both charge
quantity and specific surface area (Hou et al., 2009). Therefore, the size effect of zeta
potential of Lou soil and cinnamon soil colloidal particles is mainly related to the
reduction of charge density caused by larger specific surface area of nanoparticles (Xu
et al. 2020b).
**3.4  Aggregation kinetics curves of Lou soil and cinnamon soil colloids in NaCl and**

**CaCl$_2$ solutions**

The aggregation kinetics of Lou soil and cinnamon soil colloids in NaCl and CaCl$_2$
solutions are shown in Figs. S1 and S2. The aggregation process of soil colloids was
divided into RLA and DLA stages. The RLA stages for Lou soil colloids of $d < 2$ μm,
$d < 1$ μm and $d < 100$ nm in NaCl solution were 0–80, 0–80 and 0–100 mmol·L$^{-1}$,
respectively, during which repulsive forces existed between the particles and
attachment did not occur on every collision. As the electrolyte concentration continued
to increase, the solution entered into the DLA regime. At this point, attachment occurred
with every collision between particles, and the aggregation rates were not affected by
the electrolyte concentration. At last, the effective diameters of the formed clusters were
stable at around 1600 nm. Figure S1b, d and f showed that the aggregation behaviors
of Lou soil colloids in CaCl$_2$ solution were similar to that in NaCl solution, and the
corresponding CaCl$_2$ concentrations for Lou soil colloids of $d < 2$ μm, $d < 1$ μm and $d$



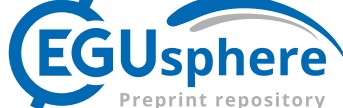

< 100 nm in RLA stage were about 0–1.5, 0–1.5 and 0–2 mmol·L$^{-1}$, respectively.
The aggregation kinetics of cinnamon soil colloids in NaCl and CaCl$_2$ solutions
were similar to Lou soil colloids (Fig. S2). The RLA stages for cinnamon soil colloids
of $d$ < 2 μm, $d$ <1 μm and $d$ <100 nm in NaCl solution were 0–100, 0–120 and 0–250
mmol·L$^{-1}$, and were about 0–1.8、0–1.7 and 0–2 mmol·L$^{-1}$ in CaCl$_2$ solution,
respectively. The effective diameters of the clusters for cinnamon soil colloids were
stabilized at about 1600 nm and 1800 nm in NaCl and CaCl$_2$ solutions, respectively.
Aggregation rates of soil colloids varied with particle diameters at the same
electrolyte concentration, which was particularly evident in RLA stage (Table 4). For
Lou soil colloids of $d$ < 2 μm, $d$ < 1 μm and $d$ < 100 nm with addition of 50 mmol·L$^{-1}$
NaCl, aggregation rates were 19.46, 14.91 and 7.22 nm·min$^{-1}$, respectively, while those
of cinnamon soil colloids were 8.98, 7.15 and 3.95 nm·min$^{-1}$ with decreasing particle
diameter. In 1 mmol·L$^{-1}$ CaCl$_2$ solution, the aggregation rates of cinnamon soil colloids
of $d$ < 2 μm, $d$ < 1 μm and $d$ < 100 nm were 8.22, 7.33 and 5.22 nm·min$^{-1}$, respectively.
Therefore, from table 4, the aggregation rates of Lou soil and cinnamon soil colloids
showed the size effect. From table 4, it could be observed that the fractal dimensions in
NaCl solutions were largely higher than those in CaCl$_2$ solutions, suggesting a much
denser structure (Meng et al., 2013). In other words, the formed structures in divalent
solutions were more open.

(*Insert Table 4 near here*)



**3.5 Suspension stability of Lou soil and cinnamon soil colloids in NaCl and CaCl$_2$ solutions**

The CCCs for Lou soil colloids of $d < 2$ μm, $d < 1$ μm and $d < 100$ nm in NaCl solution were 80.40, 91.78 and 134.96 mmol·L$^{-1}$, respectively (Fig. 3a), and those for cinnamon soil colloids were 121.10, 126.50 and 292.86 mmol·L$^{-1}$, respectively (Fig. 3b). The CCCs increased with the decreasing particle diameter, indicating that the suspension stability of soil nanoparticles was stronger than those of colloidal particles.

(*Insert Figure 3 near here*)

In CaCl$_2$ solutions, the CCCs for Lou soil colloids of $d < 2$ μm, $d < 1$ μm and $d < 100$ nm were 1.61, 1.68 and 1.77 mmol·L$^{-1}$, respectively, and for cinnamon soil colloids, those corresponding values were 1.90, 1.91 and 2.13 mmol·L$^{-1}$ (Fig. 4). The CCCs in CaCl$_2$ solutions also increased with the decreasing particle size. The contents of K$^+$, Na$^+$, Ca$^{2+}$ and Mg$^{2+}$ in Lou soil and cinnamon soil colloidal suspensions decreased with the decreasing colloidal particle diameter (Table S3), which was mainly due to the dilution effect during the extraction process. Furthermore, Table S3 showed that the soluble cation contents were rather low, and their effects on the CCCs of soil colloids could be neglected.

(*Insert Figure 4 near here*)

Based on Figs. 3 and 4, 3 mmol·L$^{-1}$ CaCl$_2$ solution could cause fast aggregation of soil colloidal particles, while it required at least 80 mmol·L$^{-1}$ NaCl solution for comparable aggregation rate, indicating that the shielding effect of divalent cations on negative charges of colloids was stronger than that of monovalent cations. The



quantitative calculation results showed that the CCC ratios of monovalent ion and
divalent ion system were between 25.64 and 27.09, which conformed to the Schulze-
Hardy rule (Baalousha, 2017).
For each type of the soil colloids, the higher the absolute zeta potential values of
colloidal particles, the more negative charges carried on the surface, and the higher the
stability (CCCs) of suspension. For the same particle diameter, e.g. $d < 100$ nm, the
absolute zeta potentials of Lou soil colloids were larger (Fig. 2) while the corresponding
CCC was lower (Figs. 3 and 4). Study on the stability of biochar nanoparticles showed
that the absolute values of zeta potentials could not be used to directly explain the
stability difference among biochar nanoparticles from different feedstock materials but
could explain the influences of solution conditions on the stability of biochar
nanoparticles derived from the same feedstock material (Xu et al., 2020a).
The CCCs of Lou soil and cinnamon soil colloids increased with decreasing
diameter; that is, the CCCs of Lou soil and cinnamon soil colloids both showed the size
effects. Hsu and Kuo (1995) demonstrated that the CCCs would generally decrease with
the increasing particle diameter because smaller particles possess thicker double
electric layers and higher electrolyte concentration is needed to neutralize charges on
the surface, which were consistent with the results of Lou soil and cinnamon soil
colloids. The above explanation by Hsu and Kuo (1995) was derived from homogenous
particles whose composition does not change with particle diameter. The results of this
paper show that, for those two alkaline soils being such heterogeneous materials, when
the organic matter contents and mineral types changed with colloidal particle diameter,



the CCCs in monovalent and divalent solutions also decreased with increasing particle
diameter.
In this paper, the organic matter contents of soil nanoparticles were the highest, so
the CCCs were the largest, which were 1.7 and 2.4 times of the corresponding colloidal
particles of $d < 2$ μm. The suspension stability of different clay minerals has been
reported to vary with the mineralogical structure. The CCC of illite ($\approx 100$ mM) in
NaCl solution was significantly higher than that of kaolinite ($\approx 20$ mM) (Jiang et al.,
2012; Xu et al., 2017). Another possible reason for the higher stability of soil
nanoparticles is the increase of illite content and the decrease of kaolinite content.
Therefore, the differences in organic matter contents and clay mineralogy are the
fundamental reasons for the differences in colloidal suspension stability behind the size
effects of Lou soil and cinnamon soil colloids.

## 4. Conclusion

The size effect of heterogeneous soil colloidal particles was demonstrated. The
number-weighted diameters for Lou soil and cinnamon soil colloids of $d < 100$ nm were
72.47 nm and 85.48 nm, respectively. With the decreasing colloidal particle diameter,
the total carbon content, organic carbon, organic functional groups content and illite all
increased, indicating the finer particle size of the organic faction and illite. The absolute
zeta potential values and the charge variability decreased with the decreasing particle
diameter. Aggregation rates of soil colloids decreased with the decreasing of particle
diameter at the same electrolyte concentration, which was particularly evident in RLA
stage. In NaCl or $CaCl_2$ solution, the CCCs increased with the decreasing Lou soil and



395 cinnamon soil colloidal particle diameter, indicating that the suspension stability was

396 enhanced. These findings have important implications for predicting the environmental

397 behaviors of soil colloids and related colloid-facilitated nutrient/contaminant transport.


## Acknowledgments


400 This work was supported by Natural Science Foundation of Shaanxi Province (2023-

401 JC-YB-263) and the National Natural Science Foundation of China (41701261), and

402 the Fundamental Research Funds for the Central Universities (2452020165).


## Author Contributions


405  Conceptualization, Xu, C.Y., Geng, Z.C., and Hu, F.N.; methodology, Xu, C.Y.;

406 software, Yan, Y.Y.; formal analysis, Liu, J.J.; investigation, Zhang, X.R.; resources,

407 Yan, Y.Y.; writing—original draft, Yan, Y.Y.; writing—review and editing, Xu, C.Y.,

408 and Hu, F.N.; visualization, Xu, C.Y., and Yan, Y.Y.; funding acquisition, Xu, C.Y.,

409 Geng, Z.C. and Hu, F.N.. All authors have read and agreed to the published version of

410 the manuscript.


## Declaration of Interest Statement


413 The authors declare that they have no known competing financial interests or personal

414 relationships that could have appeared to influence the work reported in this paper.



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





**Table captions**

**Table 1** The average diameters and distribution patterns of Lou soil and cinnamon soil colloids.

**Table 2** The physiochemical properties of Lou soil and cinnamon soil colloids.

**Table 3** The dominant clay minerals of Lou soil and cinnamon soil colloidal fractions (shown

in mass fraction, %).

**Table 4** The aggregation rates of Lou soil and cinnamon soil colloids.



**Table 1** The average diameters and distribution patterns of Lou soil and cinnamon soil
colloids

| Soil colloids | Colloidal fractions | Number-weighted diameter (nm) | Intensity-weighted diameter (nm) | $D_{10}$ (nm) | $D_{90}$ (nm) |
|---|---|---|---|---|---|
| Lou soil colloids | $d < 2$ μm | 133.16 | 396.81 | 71.53 | 232.49 |
| | $d < 1$ μm | 127.84 | 371.45 | 67.64 | 219.87 |
| | $d < 100$ nm | 72.47 | 294.10 | 38.74 | 136.72 |
| cinnamon soil colloids | $d < 2$ μm | 141.23 | 439.20 | 78.29 | 244.97 |
| | $d < 1$ μm | 131.67 | 372.07 | 75.84 | 231.64 |
| | $d < 100$ nm | 85.48 | 312.25 | 47.84 | 158.99 |

Note: $D_{10}$, $D_{50}$ and $D_{90}$ represent diameter of particles with a cumulative distribution of 10%, 50% and
90%, respectively.



**Table 2** The physiochemical properties of Lou soil and cinnamon soil colloids

| Soil colloids | Colloidal fractions | Yield (%) | Total carbon content (g·kg$^{-1}$) | Organic carbon content (g·kg$^{-1}$) | CaCO$_3$ content (g·kg$^{-1}$) | Specific surface area (m$^2$·g$^{-1}$) |
|---|---|---|---|---|---|---|
| Lou soil colloids | $d < 2$ μm | 25.12 | 20.9 | 10.9 | 10.0 | 65.37 |
| | $d < 1$ μm | 18.76 | 20.7 | 10.9 | 9.8 | 72.99 |
| | $d < 100$ nm | 6.32 | 58.2 | 27.4 | 30.8 | 45.28 |
| cinnamon soil colloids | $d < 2$ μm | 23.17 | 24.0 | 11.7 | 12.3 | 49.99 |
| | $d < 1$ μm | 16.20 | 22.3 | 12.8 | 9.5 | 61.88 |
| | $d < 100$ nm | 4.70 | 76.3 | 28.3 | 48.0 | 34.53 |




**Table 3** The dominant clay minerals of Lou soil and cinnamon soil colloidal fractions (shown

in mass fraction, %)

| Soil colloids | Colloidal fractions | Illite | Kaolinite | Chlorite | Vermiculite |
|---|---|---|---|---|---|
| Lou soil colloids | $d < 2\ \mu m$ | 34 | 23 | 4 | 9 |
|  | $d < 1\ \mu m$ | 30 | 22 | 7 | 11 |
|  | $d < 100\ nm$ | 37 | 14 | 16 | 3 |
| Cinnamon soil colloids | $d < 2\ \mu m$ | 24 | 22 | 29 | 16 |
|  | $d < 1\ \mu m$ | 31 | 19 | 25 | 12 |
|  | $d < 100\ nm$ | 37 | 16 | 17 | 5 |




**Table 4** The aggregation rates of Lou soil and cinnamon soil colloids

| Soil colloids | Colloidal fractions | Aggregation rate | | Fractal dimension | |
|---|---|---|---|---|---|
| | | In 50 mmol·L$^{-1}$ NaCl (nm·min$^{-1}$) | In 1 mmol·L$^{-1}$ CaCl$_2$ (nm·min$^{-1}$) | Na | Ca |
| Lou soil colloids | $d < 2$ μm | 19.46 | 12.01 | 1.69 ± 0.19 | 1.33 ± 0.26 |
| | $d < 1$ μm | 14.91 | 11.48 | 1.75 ± 0.06 | 1.52 ± 0.19 |
| | $d < 100$ nm | 7.72 | 9.97 | 1.71 ± 0.26 | 1.68 ± 0.13 |
| Cinnamon soil colloids | $d < 2$ μm | 8.98 | 8.22 | 1.30 ± 0.17 | 1.36 ± 0.17 |
| | $d < 1$ μm | 7.15 | 7.33 | 1.71 ± 0.24 | 1.30 ± 0.31 |
| | $d < 100$ nm | 3.95 | 5.22 | 1.52 ± 0.22 | 1.58 ± 0.19 |




**Figure captions**
**Fig. 1.** The photoelectron spectrum C1s and O1s peak diagram of Lou soil and
cinnamon soil colloids. C1s of Lou soil colloids, a. $d < 2$ μm, b. $d < 1$ μm, c. $d < 100$
nm; C1s of cinnamon soil colloids, d. $d < 2$ μm, e. $d < 1$ μm, f. $d < 100$ nm; O1s of Lou
soil colloids, g. $d < 2$ μm, h. $d < 1$ μm, i. $d < 100$ nm; O1s of cinnamon soil colloids, j.
$d < 2$ μm, k. $d < 1$ μm, l. $d < 100$ nm.
**Fig. 2.** The zeta potential of Lou soil (a) and cinnamon soil (b) colloids of $d < 2$ μm, $<$
1 μm, and $< 100$ nm at different pH
**Fig. 3.** The CCCs of Lou soil (a) and cinnamon soil (b) colloids of $d < 2$ μm, $< 1$ μm,
and $< 100$ nm in NaCl solution
**Fig. 4.** The CCC of Lou soil (a) and cinnamon soil (b) colloids of $d < 2$ μm, $< 1$ μm,
and $< 100$ nm in CaCl2 solution

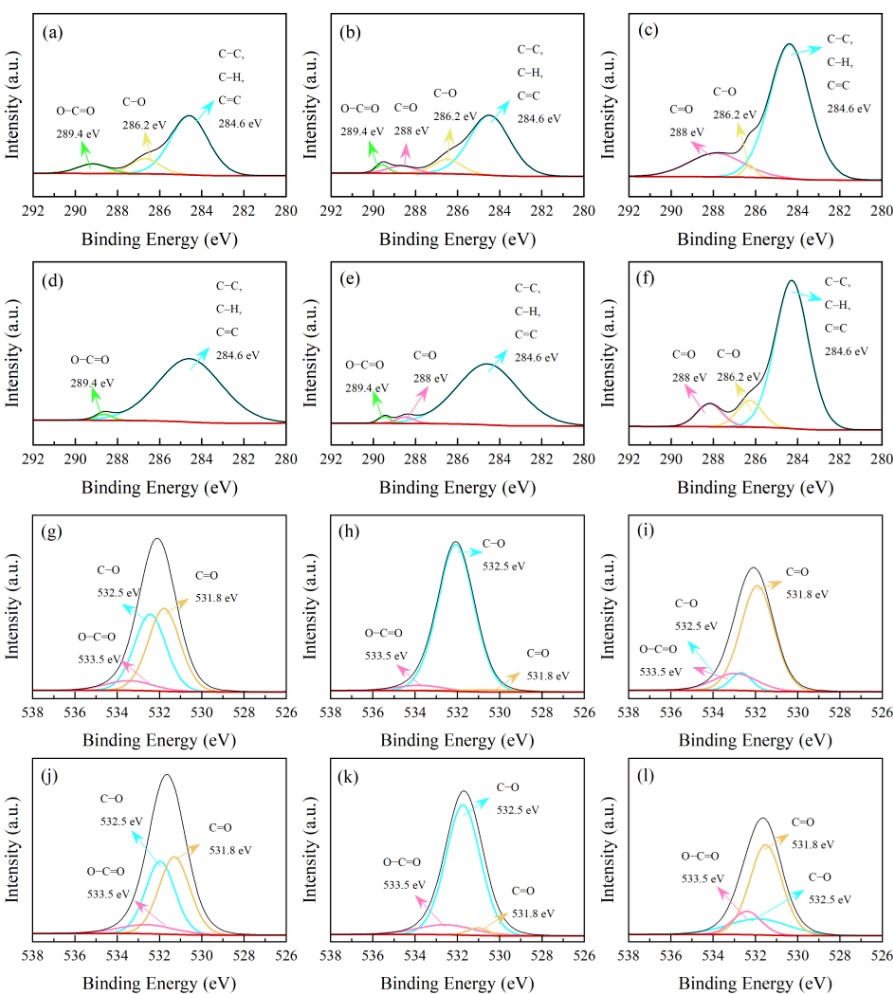

**Fig. 1** The photoelectron spectrum C1s and O1s peak diagram of Lou soil and cinnamon soil colloids. C1s of Lou soil colloids, (a). $d < 2$ μm, (b). $d < 1$ μm, (c). $d < 100$ nm; C1s of cinnamon soil colloids, (d). $d < 2$ μm, (e). $d < 1$ μm, (f). $d < 100$ nm; O1s of Lou soil colloids, (g). $d < 2$ μm, (h). $d < 1$ μm, (i). $d < 100$ nm; O1s of cinnamon soil colloids, (j). $d < 2$ μm, (k). $d < 1$ μm, (l). $d < 100$ nm.



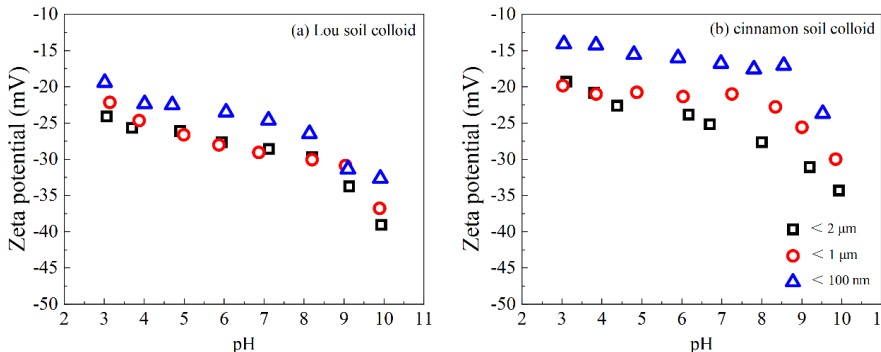


**Fig. 2** The zeta potential of Lou soil (a) and cinnamon soil (b) colloids of $d < 2$ μm, $< 1$ μm,
and $< 100$ nm at different pH





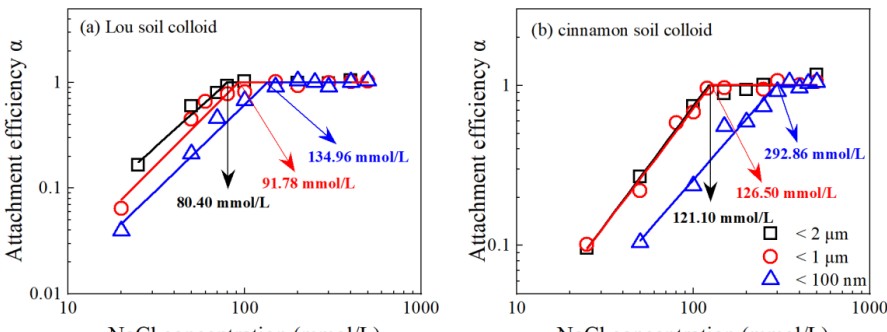


**Fig. 3** The CCCs of Lou soil (a) and cinnamon soil (b) colloids of $d$ < 2 μm, < 1 μm, and <
100 nm in NaCl solution



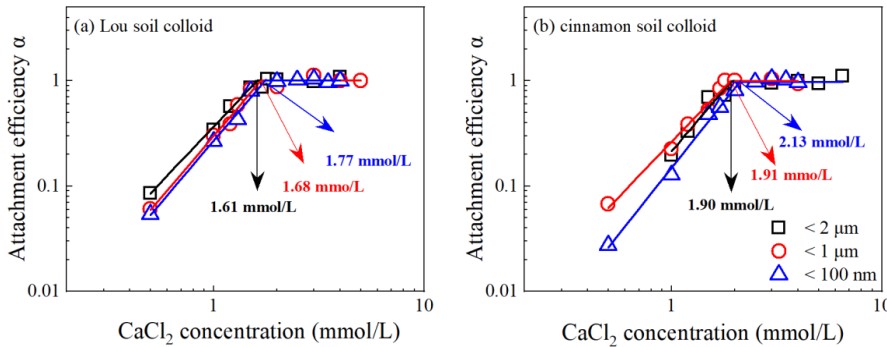

**Fig. 4** The CCC of Lou soil (a) and cinnamon soil (b) colloids of $d$ < 2 μm, < 1 μm, and < 100 nm in CaCl$_2$ solution。