# Peer review of "Effect of colloidal particle size on physicochemical properties"

_EGUsphere, 2024_

## Author Comment (AC1)

**Responses to comments from reviewers**

**Ref:** egusphere-2024-1266

**Title:** Effect of colloidal particle size on physicochemical properties and aggregation behaviors of two alkaline soils

**Journal:** SOIL

**Reviewers' comments:**

**Reviewer 1:**

The manuscript by Yan et al. presents findings on the characterization of different colloid particle size fractions extracted from calcareous soils in terms of organic and inorganic C contents and surface functional groups, clay mineralogy, and zeta potential, as well as their implications on aggregation kinetics and critical coagulation concentrations. The manuscript is generally well-written and the objectives of the study are noteworthy particularly considering the lack of information on the colloid characteristics and functioning in calcareous soils. However, the potential novelty of this work was not fully achieved due to a number of critical issues that preclude publication.

Reply: Thank you for your valuable comments on our manuscript. Please find the following detailed responses to your comments and suggestions.

1. No justification was provided for the selection of the two soils (Anthrosol and Calcisol) used for the extraction of colloid particle size fractions and the findings reported were not used to discuss any differences between the two soils. Although the fact that these soils were calcareous could represent the main novelty of this work, the knowledge gaps were not clearly identified and no hypothesis were defined.

Reply: The study selected cinnamon soil (Calcisols) as a typical calcareous soil. Calcareous soils are widely distributed in northern China and in regions worldwide with similar climatic conditions (warm temperate zone, annual average temperature of 9–14 °C, and precipitation of 600–800 mm). According to the soil genesis and formation

theory, Lou soil (Anthrosols) is formed on the basis of cinnamon soil through long-term anthropogenic maturation. In this study, both types of soil developed from loess parent material. The typical soil profile configuration for cinnamon soil is Ah-Bt-Bk-C, while for Lou soil it is Ap1-Ap2-Bt-Bk-C. According to the Chinese soil genesis classification, Lou soil is a subgroup of cinnamon soil, both being calcareous soils. This study selected two representative calcareous soils to verify the following scientific hypothesis: soil colloids are organic-inorganic composites. As particle diameter decreases (from colloid particles to nanoparticles), the number of organic functional groups on the surface of soil colloids increases, and the type of clay minerals shifts towards finer clay particles, e.g. illite, resulting in increased specific surface area and decreased charge density, and thus enhanced suspension stability, meaning particle diameter influences the composition of soil colloidal fractions, thereby changing surface properties and suspension stability.

We have re-described the scientific hypothesis of the study. Please see the revised manuscript.

2. The experimental description does not mention the presence of replicates and no uncertainties associated with the reported data have been provided making it difficult to appreciate the significance of the differences discussed.

Reply: The particle diameter reported in the study is the averaged result of 15 measurements. Total carbon and organic carbon are the averaged results of 3 measurements each. The zeta potential is the averaged result of 6 replicates. The above results are expressed in the revised manuscript as the mean ± standard error.

The critical coagulation concentration is based on a single measurement result, for the following reasons. Critical coagulation concentration (CCC) is defined as the intersection point of slow aggregation and fast aggregation, which is also a turning point of attachment efficiency *vs.* electrolyte concentration. Since the attachment efficiency is changing continually with increasing electrolyte concentration until reaching CCC, usually the aggregation curve is determined only for one time without repetition; this is generally adopted by most researches (Chen and Elimelech, 2006; Mashayekhi et al.,

2012; Zhu et al., 2014; Liu et al., 2018).

**References:**

Chen, K.L., Elimelech, M., 2006. Aggregation and deposition kinetics of Fullerene ($C_{60}$) nanoparticles. Langmuir 22: 10994–11001.

Liu, G., Zheng, H., Jiang, Z., Zhao, J., Wang, Z., Pan, B., Xing, B., 2018. Formation and physicochemical characteristics of nano biochar: insight into chemical and colloidal stability. Environ. Sci. Technol. 52(18): 10369–10379.

Mashayekhi, H., Ghosh, S., Du, P., Xing, B., 2012. Effect of natural organic matter on aggregation behavior of C60 fullerene in water. J. Colloid Interf. Sci. 374(1): 111–117.

Zhu, X., Chen, H., Li, W., He, Y., Brookes, P.C., Xu, J., 2014. Aggregation kinetics of natural soil nanoparticles in different electrolytes. Eur. J. Soil Sci. 65(2): 206–217.

3. Although the authors set out to separate (by centrifugation) and characterize three colloid particle size fractions of < 2 µm, < 1 µm and < 100 nm, the actual colloid particle diameters reported in Table 1 were very different from the intended cutoffs suggesting that the separation method used was neither calibrated nor checked beforehand (see comments below for possible causes of this).

Reply: The soil colloids of $d$ < 2 µm, < 1 µm, and < 100 nm correspond to the ranges of 1–2000 nm, 1–1000 nm, and 1–100 nm, respectively. This indicates that these components mainly consist of fine soil particles, which are similar to the findings of other researchers. The $d$ < 1 µm fraction of manual loessial soil (Anthrosol), cultivated loessial soil (Cambisol) and aeolian sandy soil (Arenosol) were extracted following the static sedimentation method, the average diameter measured by AFM were 161.161, 96.683 and 181.297 nm (Luo et al., 2017).

**References:**

Luo, X.H., Yu, L., Wang, C.Z., Yin, X.Q., Mosa, A., Lv, J.L., Sun, H.M., 2017. Sorption of vanadium (V) onto natural soil colloids under various solution pH and ionic strength conditions. Chemosphere 169: 609–617.

4. XRD spectroscopy is not quantitative (Table 3) due to the different X-ray absorption of the different minerals leading to different peak intensities. If semi-quantitative analysis was actually performed these methods must be detailed.

Reply: The XRD results are semi-quantitative analysis, which has been explained in the manuscript. The "$K$" value method is a semi-quantitative technique employed for

estimating the mineral content within soil. It operates by comparing the intensity of the dominant X-ray diffraction peak of the soil mineral colloid to a standard mineral reference (Database ICDD 2004), the relative percentage content of the minerals was determined.

**References:**

Yu, C.X., Peng, B., Tang, X.Y., Xie, S.R., Yang, G., Yin, C.Y., Tu, X.L., Liu, Q., Kesu, Y., 2009. Geochemical characteristics of soils derived from the lower-Cambrian black shales distributed in central Hunan, China. Acta Pedologica Sinica. 46(04): 557–570.

5. The authors often refer to different soil types using what seems like a national classification systems (e.g. red, yellow, purple, cinnamon; see Pg 6) that however have little significance when publishing in international journals. It would be more appropriate to use international classification systems (e.g. FAO's WRB) that also give an indication of the main soil forming processes involved.

Reply: We have updated the classification information of the soils described in the manuscript based on the World Reference Base for Soil Resources (WRB, IUSS Working Group WRB. 2022). The specific information is also presented below.

Lou soil—Anthrosols

Cinnamon soil—Calcisols

Red soil—Ferralsols

Yellow soil—Lixisols

Purple soil—Leptosols

**Reference:**

IUSS Working Group WRB. 2022. World Reference Base for Soil Resources. International soil classification system for naming soils and creating legends for soil maps. 4th edition. International Union of Soil Sciences (IUSS), Vienna, Austria.

Specific issues:

L51: Replace "active parts" with "reactive fractions"

Reply: Changes have been made according to this suggestion. The sentence has been changed "into among which soil colloids are the most reactive fractions".

L52: Surface charge is also an important characteristic of colloidal material

Reply: The sentence has been changed into "Soil colloids are characterized by high surface area and abundant surface charges, exhibiting high potential for carbon sequestration and strong adsorption capacity (Calabi-Floody et al., 2011)." according to your suggestion.

**Reference:**

Calabi-Floody, M., Bendall, J.S., Jara, A.A., Welland, M.E., Theng, B.K.G., Rumpel, C., Mora, M.L., 2011. Nanoclays from an Andisol: Extraction, properties and carbon stabilization. Geoderma 161(3–4): 159–167.

L53: What about their role in soil C mobilization and stability

Reply: We have revised the sentence based on this suggestion. The sentence has been changed into "Soil colloids are characterized by high surface area and abundant surface charges, exhibiting high potential for carbon sequestration and strong adsorption capacity (Calabi-Floody et al., 2011)." according to your suggestion.

**Reference:**

Calabi-Floody, M., Bendall, J.S., Jara, A.A., Welland, M.E., Theng, B.K.G., Rumpel, C., Mora, M.L., 2011. Nanoclays from an Andisol: Extraction, properties and carbon stabilization. Geoderma 161(3–4): 159–167.

L58: Replace "driving attached" with "mobilizing bound"

Reply: Thank you for your suggestion and corresponding changes have been made.

The sentence has been changed into "The capacity of soil colloids in mobilizing bound nutrients and pollutants is closely related to their dispersion stability under various environmental conditions" according to your suggestion.

L93: Please check this citation. I would have expected the opposite.

Reply: We revisited the original literature and verified the accuracy of our statement.

The original texts are as follows. "*With decreasing particle sizes, the two clays showed a decrease in $K_2O$, $Na_2O$, $MgO$ and $SiO_2$ contents and an increase in $Al_2O_3$ content, with the minimum molar ratio of $SiO_2$ to $Al_2O_3$ observed in nanoparticles (25-100 nm). Meanwhile, the smectite and vermiculite contents decreased or disappeared, leaving illite and kaolinite (and kaolinite interstratified minerals) as the main clay minerals in the two soil nanoparticles.*"

**Refercece:**

Zhang, Z.Y., Huang, L., Liu, F., Wang, M.K., Fu, Q.L., Zhu, J., 2016. Characteristics of clay minerals in soil particles of two Alfisols in China. Appl. Clay Sci. 120: 51–60.

L104: "The findings can have important implications"

Reply: Changes have been made according to your suggestion.

L109ff: These soils and the processes driving their formation must be clearly described. It is not clear why they have been chosen for this specific study. Other characteristics like texture and cation exchange capacity would have been important to report. What do you mean by free Fe/Al oxides? How were these quantified? It would be better to use more consolidated terms like "pedogenetic", "oxalate-extractable", "dithionite-citrate-bicarbonate extractable" etc.

Reply: The study selected cinnamon soil (Calcisols) as a typical calcareous soil. Calcareous soils are widely distributed in northern China and in regions worldwide with similar climatic conditions (warm temperate zone, annual average temperature of 9–14 °C, and precipitation of 600–800 mm). According to the soil genesis and formation theory, Lou soil (Anthrosols) is formed on the basis of cinnamon soil through long-term anthropogenic maturation. In this study, both types of soil developed from loess parent material. The typical soil profile configuration for cinnamon soil is Ah-Bt-Bk-C, while for Lou soil it is Ap1-Ap2-Bt-Bk-C. According to the Chinese soil genesis classification, Lou soil is a subgroup of cinnamon soil, both being calcareous soils.

The texture and cation exchange capacity (CEC) of soils has been supplemented in the manuscript. The CEC of Lou soil and Cinnamon soil were 25.9 cmol·kg$^{-1}$ and

22.2 cmol·kg$^{-1}$. The proportions of sand (2–0.02 mm), silt (0.02–0.002 mm) and clay (< 0.002 mm) in Lou soil were 34.0%, 40.6% and 25.4% while those were 28.0%, 44.8% and 27.2% for the cinnamon soil. The free Fe/Al oxides were extracted by Dithionite-citrate-bicarbonate (DCB) solution. We have added this information in the revised manuscript.

L123ff: Use past tense

Reply: Changes have been made according to this suggestion.

L129: "sieved". Please indicate pore size of the sieve rather than mesh number.

Reply: Changes have been made according to this suggestion. The 300-mesh sieve means sieve of a pore size of 53 μm.

L138: When using Stokes' law the density of the soil particles is normally taken to be 2.65 and not 1.65 g cm$^{-3}$. This could have been the cause for the lack of separation of the nominal colloid particle sizes fractions the authors initially planned to achieve.

Reply: Here the soil density is indeed 2.65 g·cm$^{-3}$. $\Delta d$ (1.65 g·cm$^{-3}$) is the difference in density between the soil particles (2.65 g·cm$^{-3}$) and water (1 g·cm$^{-3}$). We have added this information in the revised manuscript.

L160-161: What does this sentence mean?

Reply: Dust particles may interfere with the determination of the diameter of soil colloidal particles, so it is emphasized here to maintain the cleanliness of the instrument's operating environment. To avoid any ambiguity, we have removed this sentence.

L196ff and elsewhere: Do not report data in the main text that has already been reported in the tables and figures.

Reply: Changes have been made according to this suggestion.

L207ff: Please reword this sentence as the real meaning is not clear.

Reply: We have rephased the sentence for better understanding.

The sentence has been changed into "Given that particle diameter is proportional to the sixth power of light intensity. Consequently, in polydisperse systems where larger particles were present, the number-weighted diameter provided typically a more accurate representation of the true diameter of colloidal particles (Xu et al., 2015)".

**Reference:**

Xu, C.Y., Deng, K.Y., Li, J.Y., Xu, R.K., 2015. Impact of environmental conditions on aggregation kinetics of hematite and goethite nanoparticles. J. Nanopart Res. 17: 394.

L228: I think the authors should also consider the effect of higher organic matter contents in reducing the $N_2$-BET surface area of finest colloidal fractions due to surface coverage.

Reply: Thank you for your suggestion. The effect of organic matter on reducing the surface area has been discussed.

Lou soil and cinnamon soil nanoparticles exhibited the lowest specific surface area. Organic substances adsorb relatively little inorganic nitrogen gas during determination of specific surface area (Wilson et al., 2008; Li et al., 2013), which might result in the smallest specific surface area in soil nanoparticles rich in organic carbon. However, to our knowledge, no other better method has been reported for measuring the specific surface area of natural nanoparticles.

**Refercece:**

Li, W.Y., Zhu, X.Y., He, Y., Xing, B.S., Xu, J.M., 2013. Enhancement of water solubility and mobility of phenanthrene by natural soil nanoparticles. Environ. Pollut. 176: 228–233.

Wilson, M.A., Tran, N.H., Milev, A.S., Kannangara, G.S.K., Volk, H., 2008. Nanomaterials in soils. Geoderma 146: 291–302.

L251: Are these differences really significant?

Reply: Those comparison was made based on semi-quantitative calculation. Other researchers, like Wang et al (2019), have also made the same comparison. The sentence has been changed into "High-resolution spectra of C1s and O1s of soil colloids were

acquired by X-ray photoelectron spectroscopy (XPS) (Thermo Scientific K-Alpha, USA), and the Gaussian−Lorentzian curve-fitting program (XPSPEAK 4.1) was used to analyze the XPS spectra".

**Refercece:**

Wang, Y., Zhang, W., Shang, J.Y., Shen, C.Y., Joseph, S.D., 2019. Chemical Aging Changed Aggregation Kinetics and Transport of Biochar Colloids. Environ. Sci. Technol. 53(14).

L280: When referring to zeta potential of colloidal fractions I suggest the authors refer to "less negative" and "more negative" when comparing values. For example, here "zeta potential became more negative with increasing particle diameter".

Reply: Changes have been made according to this suggestion.

L386: The conclusions should summarise the main findings considering the application of this work to understand soil functions and not merely repropose the results.

Reply: Changes have been made according to this suggestion. The sentence has been changed into "The size effect of the composition, surface properties and aggregation behaviors on particle size for heterogeneous soil colloidal particles was systematically studied. The organic carbon content of Lou soil and cinnamon soil nanoparticles was the highest, which was 27.38 g·kg$^{-1}$ and 28.31 g·kg$^{-1}$, respectively. The zeta potential became less negative and the charge variability decreased with decreasing particle diameter. The CCCs of Lou soil and cinnamon soil nanoparticles were about 1.7 and 2.4 times of the corresponding colloidal particles of $d < 2$ μm in NaCl solutions, while those were about 1.1 and 2.4 times in CaCl$_2$ solutions. With the decreasing colloidal particle diameter, variations in soil colloidal organic carbon, organic functional groups content and illite affected the zeta potential and charge variability, resulting in enhanced suspension stability. This study confirmed the size effect of heterogeneous soil colloidal particles. These findings have important implications for understanding the mechanism of size effect on soil colloid properties and environmental behavior, as well as for predicting the transport of nutrients and pollutants facilitated by these colloids".

**Reviewer 2:**

This study examined the impact of particle size on surface properties and aggregation behaviors of soil colloids, aiming to enhance the understanding of these properties across different particle sizes in natural environments. However, there are still some issues in this study that need to be clarified. What is the background and significance of selecting these two soils (Lou soil and cinnamon soil) in this study? Additionally, while the study considered colloids of varying sizes (d < 2 μm, d < 1 μm, and d < 100 μm), the final measured colloid particle size fell outside of this range (Lines 195-202). The primary objective of the research is to explore the influence of colloid particle size on colloid aggregation; however, the presence of organic matter on colloid surfaces raises questions about how the study differentiates between the effects of colloid size and organic matter on colloid aggregation. Essentially, the study is about the two soils, rather than using the two alkaline soils to explore broader and more generalizable phenomena, and lacks the hypothesis.

Reply: Thank you for your time and valuable comments.

The soil colloids of $d < 2$ μm, $< 1$ μm, and $< 100$ nm correspond to the ranges of 1–2000 nm, 1–1000 nm, and 1–100 nm, respectively. This indicates that these components mainly consist of fine soil particles, which are similar to the findings of other researchers. The $d < 1$ μm fraction of manual loessial soil (Anthrosol), cultivated loessial soil (Cambisol) and aeolian sandy soil (Arenosol) were extracted following the static sedimentation method, the average diameter measured by AFM were 161.161, 96.683 and 181.297 nm (Luo et al., 2017).

The study was designed to compare how particle size has changed both organic matter content and clay mineralogy of soil colloidal fraction. The study selected cinnamon soil (Calcisols) as a typical calcareous soil. Cinnamon soil (Calcisols) is widely distributed in northern China and in regions worldwide with similar climatic conditions (warm temperate zone, annual average temperature of 9–14 °C, and precipitation of 600–800 mm). According to the soil genesis and formation theory, Lou soil (Anthrosols) is formed on the basis of cinnamon soil through long-term anthropogenic maturation. In this study, both types of soil developed from loess parent

material. The typical soil profile configuration for cinnamon soil is Ah-Bt-Bk-C, while for Lou soil it is Ap1-Ap2-Bt-Bk-C. According to the Chinese soil genesis classification, Lou soil is a subgroup of cinnamon soil, both being calcareous soils. This study selected two representative calcareous soils to verify the following scientific hypothesis: soil colloids are organic-inorganic composites. As particle diameter decreases (from colloid particles to nanoparticles), the number of organic functional groups on the surface of soil colloids increases, the type of clay minerals shifts towards finer clay particles, e.g. illite, resulting in increased specific surface area and decreased charge density, and thus enhanced suspension stability, meaning particle diameter influences the composition of soil colloidal fractions, thereby changing surface properties and suspension stability.

**References:**

Luo, X.H., Yu, L., Wang, C.Z., Yin, X.Q., Mosa, A., Lv, J.L., Sun, H.M., 2017. Sorption of vanadium (V) onto natural soil colloids under various solution pH and ionic strength conditions. Chemosphere 169: 609–617.

Specific comments

Line 21 Delete "most" and "of all"

Reply: Changes have been made according to this suggestion. The sentence has been changed into "Soil colloidal particles are the active components, and they also vary in elemental composition and environmental behaviors with the particle size".

Line 56 change "of" to "in"?

Reply: Changes have been made according to this suggestion. The sentence has been changed into "Due to their high reactivity and fluidity in aqueous environment, soil colloids play an important role in physical, chemical and biogeochemical processes in natural environment".

Line 115 change "are" to "were"?

Reply: Changes have been made according to this suggestion. The sentence has been changed into "The basic soil properties were determined based on standard methods".

Lines 109-112: Please add detailed information on sampling points, such as longitude and latitude, crop type, climate, etc.

Reply: The study investigated two representative surface soils (0–20 cm) by 5 to 10 mixed soil samples which collected using a stainless-steel auger at Yangling District (N38°18′14″ and E108°2′30″), and Zhouzhi Country (N34°8′8″ and E108°3′10″), in Shaanxi province, northwest China, respectively. The primary crops cultivated in this region included winter wheat (*Triticum aestivum* L.), broom corn millet (*Panicum miliaceum* L.), and maize (*Zea mays* L.). Cinnamon soil (Calcisols) as a typical calcareous soil, which are widely distributed in northern China and in regions worldwide with similar climatic conditions (warm temperate zone, annual average temperature of 9–14 °C, and precipitation of 600–800 mm). According to the soil genesis and formation theory, Lou soil (Anthrosols) is formed on the basis of cinnamon soil through long-term anthropogenic maturation. In this study, both types of soil developed from loess parent material. The typical soil profile configuration for cinnamon soil is Ah-Bt-Bk-C, while for Lou soil it is Ap1-Ap2-Bt-Bk-C. According to the Chinese soil genesis classification, Lou soil is a subgroup of cinnamon soil, both being calcareous soils.

Line 114-119: Specific methods for determining soil properties need to be given.

Reply: Detailed information about soil property determination has been added based on this suggestion.

Soils samples were taken back to laboratory for air-drying and sieving. The basic soil properties were determined based on standard methods. Soil pH was measured with a pH electrode, employing a solution-to-soil ratio of 2.5:1. Soil organic carbon (SOC) was determined using the $K_2Cr_2O_7$ oxidation method. The cation exchange capacity (CEC) of soil was measured with exchange method. The $CaCO_3$ content was determined by gasometric method. The free Fe/Al oxides were extracted by dithionite-citrate-bicarbonate (DCB) solution. The particle size distribution was measured using the laser diffractometer of Malvern Mastersizer 2000 (Malvern Instruments Ltd., UK).

Line 123: 50 g of dry soil was weighed into a beaker containing 500 mL of distilled water.

Reply: Changes have been made according to this suggestion.

Line 204–207: delete "The particles in the soil solution are in constant Brownian motion, and when light passes through the colloids, the particles will undergo light scattering, resulting in fluctuations in light intensity, and thus the effective diameter (intensity-weighted diameter) of the particles is calculated (Filella et al., 1997)" and re-written it.

Reply: The sentence has been rephrased.

The colloidal particles in the soil solution were in constant Brownian motion, upon illumination by light, these colloidal particles scatter light, causing variations in light intensity. This phenomenon allowed for the calculation of the effective diameter of the particles, which was the intensity-weighted diameter (Filella et al., 1997).

Line 213 higher than

Reply: Changes have been made.

Lines 276-277: What is the reason for this phenomenon?

Reply: The zeta potential is positively proportional to surface charge density (Li and Xu, 2008). The combined determination method revealed that Lou soil possess a higher surface charge density compared to cinnamon soil, consequently resulting in a more negative zeta potential for Lou soil. We have added this information in the revised manuscript.

**Reference:**

Li, S.Z., Xu, R.K., 2008. Electrical double layers' interaction between oppositely charged particles as related to surface charge density and ionic strength. Colloids Surf. A. 326(3): 157–161.

Lines 380-381: Why does the increase of illite content increase the colloidal stability?

Reply: Previous research on mineral colloids has established that the CCC of illite ($\approx$

100 mM) in NaCl solution was significantly higher than that of kaolinite ($\approx 20$ mM) (Jiang et al., 2012; Xu et al., 2017), indicating that the stability of illite suspensions is significantly higher than that of kaolinite. In this study, with the decrease of particle size, the content of illite increased and kaolinite content decreased. Therefore, the colloidal suspension stability was enhanced. We have added this information in the revised manuscript.

**Reference:**

Jiang, C.L., Séquaris, J.M., Vereecken, H., Klumpp, E., 2012. Effects of inorganic and organic anions on the stability of illite and quartz soil colloids in Na-, Ca- and mixed Na-Ca systems. Colloids Surf. A. 415: 134–141.

Xu, C.Y., Xu, R.K., Li, J.Y., Deng, K.Y., 2017. Phosphate-induced aggregation kinetics of hematite and goethite nanoparticles. J. Soils Sediments. 17: 352–363.

Table 2: Why do the colloids with $d < 1$ µm have the largest specific surface area?

Reply: The specific surface areas for colloidal fractions of $d < 100$ nm were lower than those of $d < 1$ µm, which may be related to the structures of formed clusters while drying the samples for observation under microscopy (Yu et al., 2017; Weissenberger et al., 2021). It may also be that organic substances adsorb relatively little inorganic nitrogen (Wilson et al., 2008; Li et al., 2013). Therefore, the BET-N$_2$ method may not be suitable for soil colloids with high organic carbon content, but to our knowledge, there is no other suitable determination method. We have added this information in the revised manuscript.

**Reference:**

Yu, X., Fu, Y., Lu, S., 2017. Characterization of the pore structure and cementing substances of soil aggregates by a combination of synchrotron radiation X-ray micro–computed tomography and scanning electron microscopy. Eur. J. Soil Sci. 68(1): 66–79.

Weissenberger, G., Henderikx, R.J., Peters, P.J., 2021. Understanding the invisible hands of sample preparation for cryo-EM. Nat. Methods. 18(5): 463–471.

Wilson, M.A., Tran, N.H., Milev, A.S., Kannangara, G.S.K., Volk, H., 2008. Nanomaterials in soils. Geoderma 146: 291–302.

Li, W.Y., Zhu, X.Y., He, Y., Xing, B.S., Xu, J.M., 2013. Enhancement of water solubility and mobility of phenanthrene by natural soil nanoparticles. Environ. Pollut. 176: 228–233.

---

## Author Response (AR2)

**Responses to comments from editor**

**Ref:** egusphere-2024-1266

**Title:** Effect of colloidal particle size on physicochemical properties and aggregation behaviors of two alkaline soils

**Journal:** SOIL

**Editor' comments:**

Thank you very much for revising the manuscript and taking almost all suggestions of the reviewers into account.

Reply: Thank you for your valuable comments on our manuscript. Please find the following detailed responses to your comments and suggestions.

Firstly, I am still not convinced by the justification for the choice of the two soils. You have not answered why you chose these two soils in relation to your main hypothesis and the main processes you are investigating. What makes these soils so special that you could carry out your research?

Reply: Lou soil (Anthrosols) and cinnamon soil (Calcisols) are calcareous soils developed from loess parent materials, which are the most common and characteristic soil types on the Guanzhong Plain, Shaanxi Province, China. Among these, Anthrosols, as a unique calcareous soil, have formed on the basis of cinnamon soil through long-term anthropogenic maturation. The primary objective of our manuscript is to reveal the effects of particle diameter on the organic components, clay minerals, surface electrochemical properties, and colloidal stability of these two most predominant alkaline soils collected within Shaanxi Province, and to analyze how the particle size affects the surface properties and suspension stability by changing the composition of soil colloidal components.

However, It should be noted that cinnamon soil (Calcisols) is one of the most representative soil groups in northern China, widely distributed in temperate climatic conditions of China, including provinces such as Shaanxi, Shanxi, Shandong, Hebei,

Gansu, and Ningxia. The present study focuses on Lou soil (Anthrosols) and cinnamon soil (Calcisols). Considering the high heterogeneity in soil constitutes, which have profoundly caused the size effects of colloids, further research on other typical zonal soils will be conducted in subsequent studies.

Please add the number of replicates for each parameter analyzed. This is a very important information for the reader of the journal.

Reply: The particle diameter reported in the study is the averaged result of 15 measurements. Total carbon and organic carbon are the averaged results of 3 measurements each. The zeta potential is the averaged result of 6 replicates. The above results are expressed in the revised manuscript as the mean ± standard error.

The critical coagulation concentration is based on a single measurement result, for the following reasons. Critical coagulation concentration (CCC) is defined as the intersection point of slow aggregation and fast aggregation, which is also a turning point of attachment efficiency *vs.* electrolyte concentration. Since the attachment efficiency is changing continually with increasing electrolyte concentration until reaching CCC, usually the aggregation curve is determined only for one time without repetition; this is generally adopted by most researches (Chen and Elimelech, 2006; Mashayekhi et al., 2012; Zhu et al., 2014; Liu et al., 2018).

We have added this information in the revised manuscript.

**References:**

Chen, K.L., Elimelech, M., 2006. Aggregation and deposition kinetics of Fullerene ($C_{60}$) nanoparticles. Langmuir 22: 10994–11001.

Liu, G., Zheng, H., Jiang, Z., Zhao, J., Wang, Z., Pan, B., Xing, B., 2018. Formation and physicochemical characteristics of nano biochar: insight into chemical and colloidal stability. Environ. Sci. Technol. 52(18): 10369–10379.

Mashayekhi, H., Ghosh, S., Du, P., Xing, B., 2012. Effect of natural organic matter on aggregation behavior of C60 fullerene in water. J. Colloid Interf. Sci. 374(1): 111–117.

Zhu, X., Chen, H., Li, W., He, Y., Brookes, P.C., Xu, J., 2014. Aggregation kinetics of natural soil nanoparticles in different electrolytes. Eur. J. Soil Sci. 65(2): 206–217.